# CaV1.1 voltage-sensing domain III exclusively controls skeletal muscle excitation-contraction coupling

Simone Pelizzari [1], Martin C. Heiss [1], Monica L. Fernández-Quintero [2], Yousra El Ghaleb [1], Klaus R. Liedl [2], Petronel Tuluc [3], Marta Campiglio [1] & Bernhard E. Flucher [1]✉

Skeletal muscle contractions are initiated by action potentials, which are sensed by the voltage-gated calcium channel (CaV1.1) and are conformationally coupled to calcium release from intracellular stores. Notably, CaV1.1 contains four separate voltage-sensing domains (VSDs), which activate channel gating and excitation-contraction (EC-) coupling at different voltages and with distinct kinetics. Here we show that a single VSD of CaV1.1 controls skeletal muscle EC-coupling. Whereas mutations in VSDs I, II and IV affect the current properties but not EC-coupling, only mutations in VSD III alter the voltage-dependence of depolarization-induced calcium release. Molecular dynamics simulations reveal comprehensive, non-canonical state transitions of VSD III in response to membrane depolarization. Identifying the voltage sensor that activates EC-coupling and detecting its unique conformational changes opens the door to unraveling the downstream events linking VSD III motion to the opening of the calcium release channel, and thus resolving the signal transduction mechanism of skeletal muscle EC-coupling.

Voltage-gated calcium channels (CaV) transduce the electrical signals of excitable cells into calcium-mediated functions like the secretion of neurotransmitters in the nervous system or the contraction of the heart and skeletal muscles[1]. This fundamental function of CaVs depends on their ability to sense changes of the membrane potential and, in response, open a calcium-selective conduction pore. For this purpose, voltage-gated cation channels have four independent voltage-sensing domains (VSDs) arranged around a common channel pore (Fig. 1). Each VSD is composed of four trans-membrane helices (S1–S4). The S4 helix contains 4 to 6 regularly spaced positively charged amino acids, functioning as gating charges[2]. At resting membrane potentials, the positively charged S4 helix is pulled inward towards the cytoplasmic side. Upon depolarization, S4 moves outward. During this transition, the middle gating charges pass through the hydrophobic constriction site at which the membrane electric field

is focused. Transient formation of ionic bonds between the positive gating charges in the S4 helix and negative countercharges in the neighboring S2 and S3 helices facilitates this transition and stabilizes the VSD in the activated state[3]. The resulting conformational change in the VSDs allosterically couples to the pore domain, thus controlling the opening and closing of the channel gate.

Eukaryotic voltage-gated calcium channels are products of single genes. Their four VSDs share the prototypical voltage sensor fold but are structurally and functionally distinct from each other[4]. They respond differently to depolarization and contribute differentially to channel gating[5–7]. The skeletal muscle voltage-gated calcium channel (CaV1.1) is a striking example for such task sharing among the four VSDs of an ion channel. First, CaV1.1 serves two functions: It is the voltage-sensor of skeletal muscle excitation-contraction (EC-) coupling as well as a slowly activating L-type calcium channel[8,9]. Secondly, CaV1.1

[1]Institute of Physiology, Department of Physiology and Medical Biophysics, Medical University Innsbruck, 6020 Innsbruck, Austria. [2]Institute of General, Inorganic and Theoretical Chemistry, University of Innsbruck, Innsbruck, Austria. [3]Department of Pharmacology and Toxicology, Center for Molecular Biosciences Innsbruck, University of Innsbruck, 6020 Innsbruck, Austria. ✉e-mail: bernhard.e.flucher@i-med.ac.at

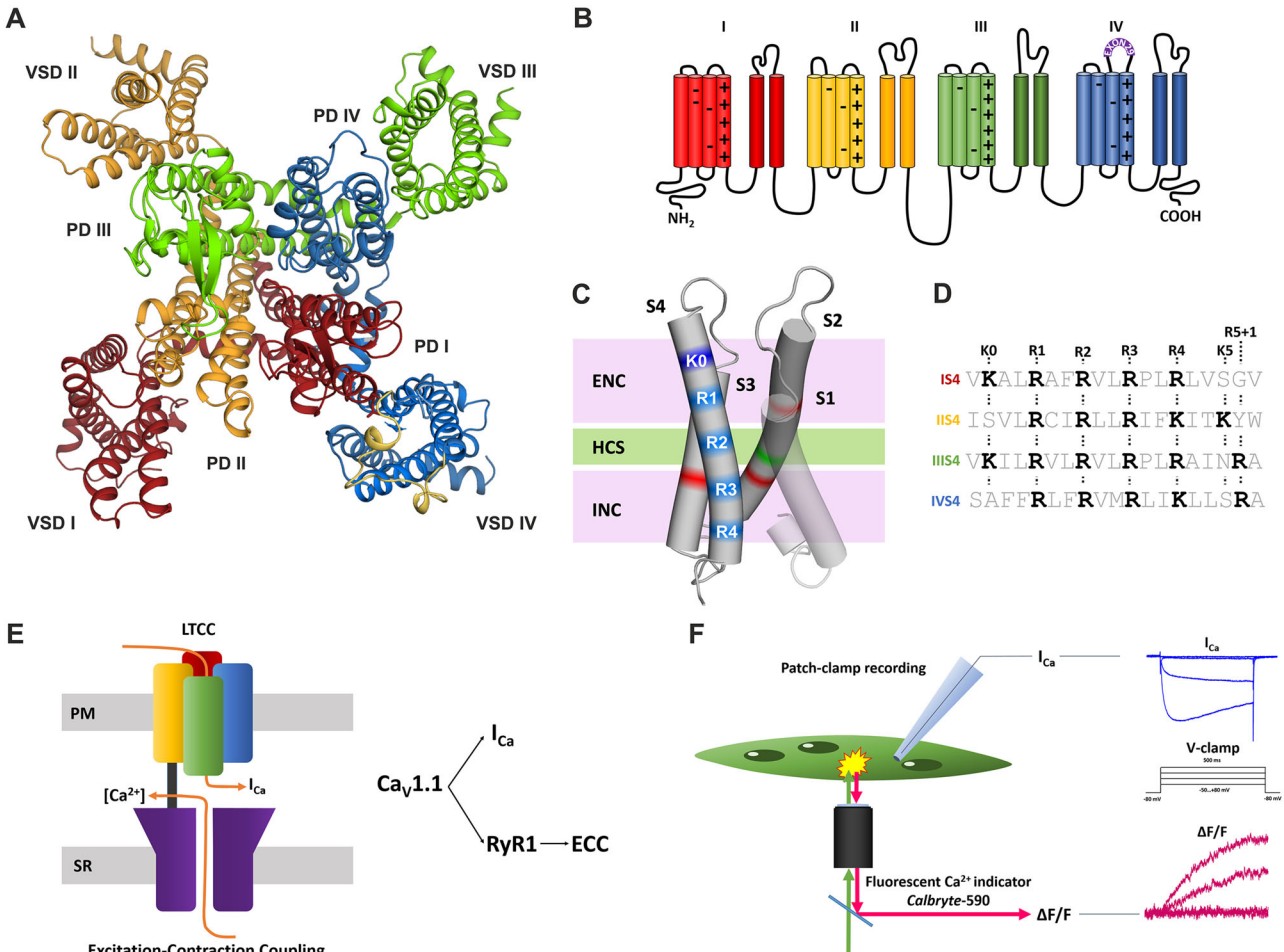

**Fig. 1 | Structure and functions of the skeletal muscle voltage-gated calcium channel Ca$_V$1.1. A** Protein structure (top view) showing the four-fold symmetry of Ca$_V$1.1 with four voltage-sensing domains (VSD) arranged around the common channel pore (PD). **B** The Ca$_V$1.1 membrane topology reveals four homologous repeats, each with six membrane helices (S1–S6) (colors as in **A**). **C** A VSD contains four transmembrane helices (S1–S4). S4 carries the regularly spaced positive gating charges (blue bands), which upon state transition successively interact with

negative countercharges in the S2 and S3 helices (red bands) of the intracellular and extracellular negative clusters (INC, ENC). HCS indicates the hydrophobic constriction site. **D** The number and type of gating charges (R or K) differ between the S4 helices of the four VSDs. **E** In skeletal muscle Ca$_V$1.1 functions as calcium channel as well as voltage sensor for EC-coupling. **F** Here we simultaneously measure calcium currents and fluorescence calcium signals (EC-coupling) in dysgenic (Ca$_V$1.1-null) myotubes, reconstituted with different Ca$_V$1.1 constructs.

controls these two voltage-dependent functions independently from each other. EC-coupling activates rapidly and at substantially lower membrane potentials than the current through Ca$_V$1.1. In fact, skeletal muscle EC-coupling is independent of the calcium influx through the channel[10]. It solely relies on calcium release from cytoplasmic stores triggered by the Ca$_V$1.1 voltage sensor. It is generally accepted that conformational changes in Ca$_V$1.1 in response to membrane depolarization are mechanically coupled to the opening of the adjacent calcium release channel, the type 1 ryanodine receptor (RyR1)[11,12]. Although the essential proteins for this signaling process have all been defined[13], the molecular mechanism by which voltage-sensing by Ca$_V$1.1 is transduced into the opening of the RyR1, as well as the specific roles of the four VSDs in skeletal muscle EC-coupling are still elusive.

Existing evidence suggests that VSDs I and IV of Ca$_V$1.1 are critical for controlling the current properties but not for EC-coupling. VSD I is the major determinant of the characteristic slow activation kinetics of Ca$_V$1.1 currents[14,15]. However, EC-coupling shows an intrinsically fast activation, suggesting that it is independent of VSD I. VSD IV determines the characteristic right-shifted voltage dependence of current activation[16–18]. Inclusion of the alternatively spliced exon 29 in the extracellular loop connecting S3 and S4 of VSD IV shifts the voltage-dependence of current activation to more positive potentials without

affecting the voltage-dependence of EC-coupling, thus indicating that activation of EC-coupling is independent of VSD IV. Two recent voltage-clamp fluorometry studies compared the voltage sensitivity and kinetics of the individual Ca$_V$1.1 VSDs with the activation properties of EC-coupling[7,19]. However, these studies disagree on the potential roles of VSDs II and III in the activation of EC-coupling.

Here we apply two complementary mutagenesis strategies to resolve this problem and demonstrate which VSDs of Ca$_V$1.1 control channel gating and which skeletal muscle EC-coupling. Mutation of the innermost gating charges of VSD III right-shifts the voltage-dependence of EC-coupling by about 45 mV and insertion of the extracellular IVS3-S4 loop into VSD III left-shifts the voltage-dependence of EC-coupling by 60 mV. The same mutations in the other three VSDs have no effect on EC-coupling. In contrast, mutations in all VSDs differentially alter the gating properties of the calcium currents. These findings demonstrate that activation of EC-coupling requires the action of only one of the four Ca$_V$1.1 VSDs (VSD III), whereas all four VSDs contribute to channel gating to different degrees. Moreover, MD simulations reveal a non-canonical movement of VSD III in response to membrane depolarization that might initiate the conformational coupling between the Ca$_V$1.1 voltage-sensor and the RyR1 calcium release channel in skeletal muscle contraction.

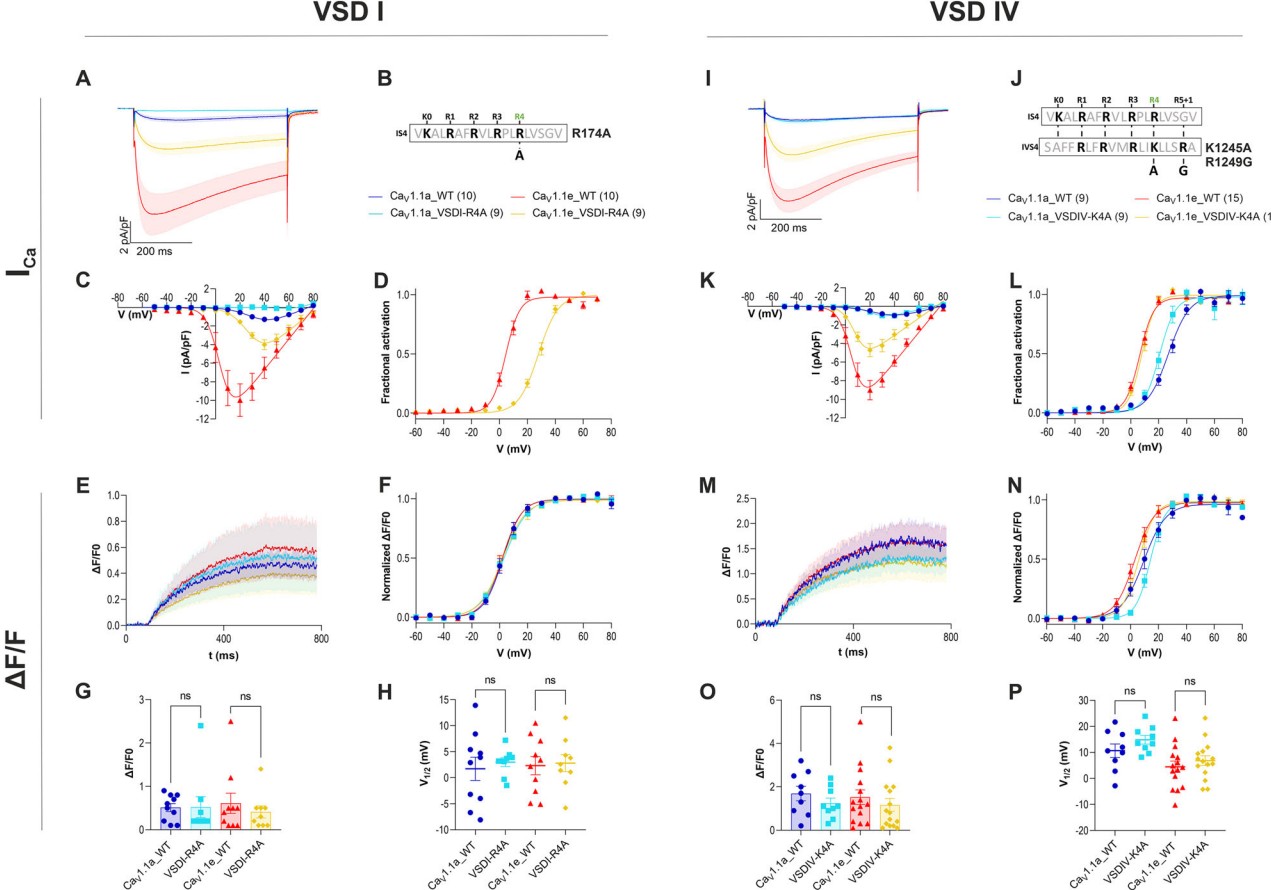

**Fig. 2 | Charge-neutralizing mutations of gating charges in VSDs I and IV, and insertion of exon 29 in VSD IV alter current properties but not EC-coupling.** **A, I** Mean ± SEM of the calcium currents in response to 500 ms depolarization to $V_{max}$ of wildtype (**A**: blue, $n = 10/N = 4$; red $n = 10/N = 6$; **I**: blue, $n = 9/N = 6$; red $n = 15/N = 6$) and mutant (**A**: turquois, $n = 9/N = 5$; yellow, $n = 9/N = 4$; **I**: turquois, $n = 9/N = 6$; yellow, $n = 15/N = 6$) Ca$_V$1.1a and Ca$_V$1.1e expressed in dysgenic myotubes. **B, J** Amino acid sequences of IS4 and IVS4 indicating the gating charges and the R174A (**B**) and K1245A, R1249G (**J**) substitutions. **C, D, K, L** I/V-curves and fractional activation curves showing the effects of the mutations on amplitudes and voltage-dependence of calcium currents. **E, M** Mean ± SEM of the cytoplasmic calcium transients of wildtype and mutant Ca$_V$1.1a and Ca$_V$1.1e constructs in response to 500 ms depolarizations to $V_{max}$. **E–H, M–P** Amplitudes and voltage-dependence of depolarization-induced calcium release remain unaffected by the substitution of the innermost gating charges in VSDs I and IV (Ca$_V$1.1a WT vs. R4A: (**G**) $p = 0.96$; (**H**) $p = 0.62$; (**O**) $p = 0.3$; (**P**) $p = 0.19$) (Ca$_V$1.1e WT vs. R4A: (**G**) $p = 0.49$; (**H**) $p = 0.84$; (**O**) $p = 0.44$; (**P**) $p = 0.42$), as well as by alternative splicing of exon 29 in VSD IV (Ca$_V$1.1a vs. Ca$_V$1.1e). Mean ± SEM; mutants and wt controls compared by two-tailed student's $t$-tests, *$\triangleq p < 0.05$,**$\triangleq p < 0.01$,***$\triangleq p < 0.001$. $n =$ number of recorded cells; $N =$ number of transfections.

# Results

## VSDs I and IV control currents only

A disease mutant of Ca$_V$1.1 affecting the innermost gating charge of VSD I, R174W, has been shown to abolish the calcium currents without affecting EC-coupling[20,21]. This finding corroborated the notion that the normal function of VSD I is essential for channel gating, but not for EC-coupling. Here, we replicated this experiment using a more subtle amino acid substitution, R174A, in the GFP-tagged rabbit skeletal muscle Ca$_V$1.1a adult isoform expressed in dysgenic (Ca$_V$1.1-null) myotubes[22–24]. Co-clustering of Ca$_V$1.1 with RyR1 confirmed normal expression and correct targeting of the recombinant channels into skeletal muscle triads, the junctions between transverse tubules or the plasma membrane and the sarcoplasmic reticulum (SR), where EC-coupling takes place (Supplementary Fig. 1A)[24]. We simultaneously analyzed calcium currents using patch-clamp recordings and depolarization-induced changes of the cytoplasmic free calcium concentration (i.e. EC-coupling) applying microfluorometric recordings of the fluorescent calcium indicator Calbryte-590 (Fig. 1F). Consistent with previous studies, expression of the wildtype Ca$_V$1.1a in dysgenic myotubes restored EC-coupling with a half-maximal activation ($V_{1/2}$) near 0 mV as well as small amplitude calcium currents at a $V_{1/2}$ near 30 mV (Supplementary Table 1). Similar to the disease mutant

R174W[20], substitution of the innermost gating charge with alanine (Ca$_V$1.1a_VSDI-R4A) completely abolished the calcium currents but did not affect the amplitude (ΔF/F) or voltage-dependence ($V_{1/2}$) of depolarization-induced calcium transients (Fig. 2). In the embryonic splice variant Ca$_V$1.1e, exclusion of the nineteen amino acids encoded by exon 29 from the extracellular IVS3-S4 loop substantially increases the current amplitude and specifically left-shifts the voltage-dependence of current activation, without affecting the amplitude and voltage-dependence of depolarization-induced calcium release. Thus, in Ca$_V$1.1e calcium currents and EC-coupling activated at about the same potential near 0 mV[16,17]. In the background of this channel variant, the R174A substitution (Ca$_V$1.1e_VSDI-R4A) reduced the current density and shifted its voltage-dependence of activation by 23 mV to more depolarized potentials compared to wildtype Ca$_V$1.1e (Fig. 2A–D, Supplementary Table 1). Again, activation of EC-coupling remained unaffected by the R174A substitution, consistent with a role of VSD I in regulating channel gating, but not EC-coupling (Fig. 2E–H).

Next, we applied the same experimental strategy to VSD IV. In VSD IV the gating charge corresponding to R174 (VSD I, R4) is K1245 (K4), but the S4 helix of this VSD contains an additional positive charge four residues downstream (R1249). Not knowing which of the two might function as the innermost gating charge, we

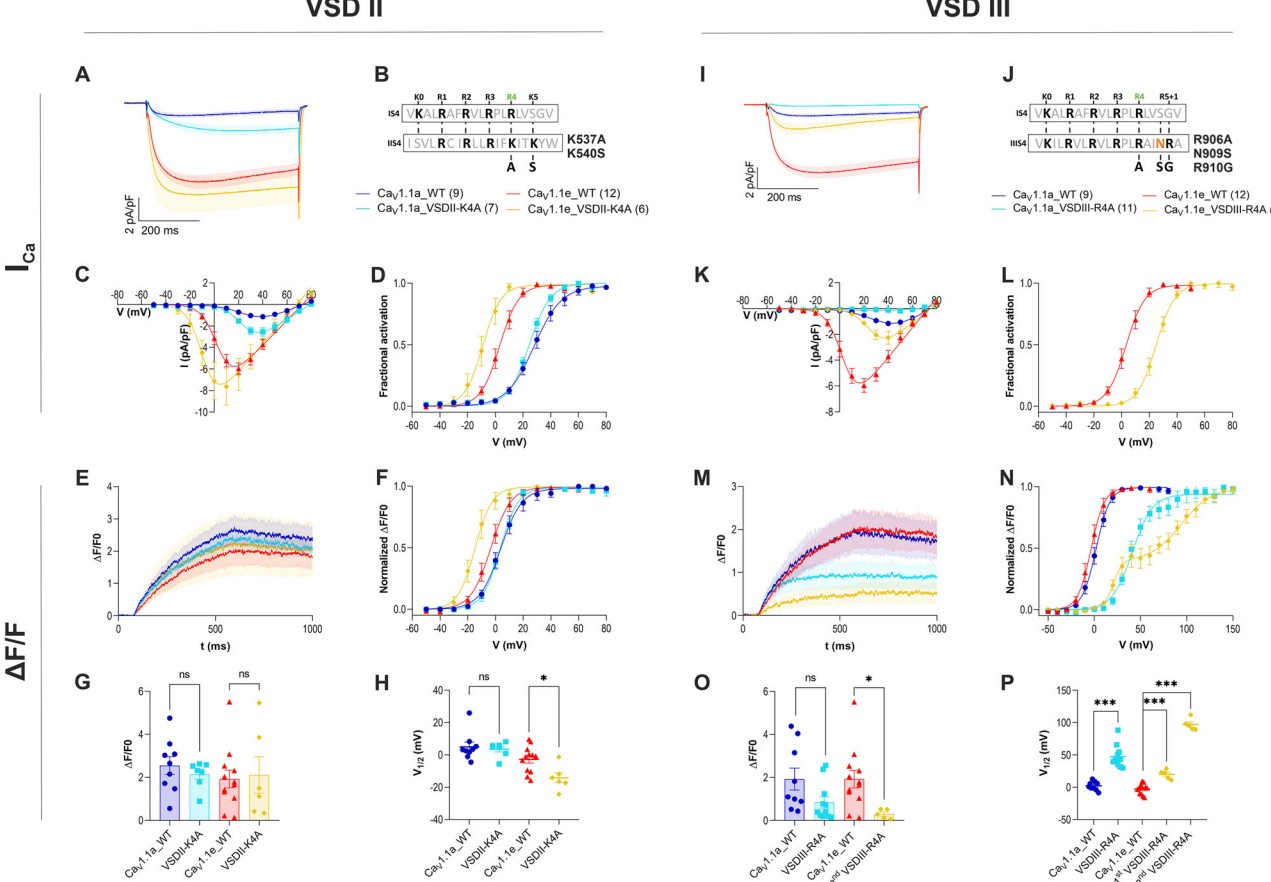

**Fig. 3 | Neutralizing the innermost gating charges in VSD II and VSD III of Ca$_V$1.1 differentially affects the properties of calcium currents and of EC-coupling.**
**B, J** In VSDs II and III the gating charges in position R4 were substituted by alanine and the amino acids in positions R5 and R5 + 1, respectively, were replaced by the corresponding residues of VSD I of Ca$_V$1.1a and Ca$_V$1.1e. **A–D** In VSD II voltage-dependence of calcium currents activation (I$_{Ca}$) was shifted to more negative potentials in Ca$_V$1.1e_VSDII-K4A (yellow $n = 6/N = 4$), but not in Ca$_V$1.1a_VSDII-K4A (turquois, $n = 7/N = 4$) compared to matched controls (red, $n = 12/N = 9$; blue, $n = 9/N = 3$). **E–H** The amplitudes of the calcium transients (ΔF/F) were similar in WT and mutants of VSD II (**G**: Ca$_V$1.1a, $p = 0.44$; Ca$_V$1.1e $p = 0.82$). Ca$_V$1.1e_VSDII-K4A shifted the V$_{1/2}$ of EC-coupling (**H**: yellow, $p = 0.011$) by the same extent as the current activation, whereas Ca$_V$1.1a_VSDII-K4A showed no effect (**H**: turquois, $p = 0.69$).

**I–L** In VSD III the amino acid substitutions abolished calcium currents in Ca$_V$1.1a_VSDIII-R4A (**I**: turquois, $n = 11/N = 4$; blue, $n = 9/N = 3$) and in Ca$_V$1.1e_VSDIII-R4A reduced its amplitude and shifted its voltage-dependence of activation to more positive potentials (red, $n = 12/N = 9$; yellow, $n = 5/N = 4$). **M–P** In both mutant constructs the depolarization-induced calcium transients (ΔF/F) were reduced in amplitude (**O**: Ca$_V$1.1a_VSDIII-R4A, $p = 0.08$; Ca$_V$1.1e_VSDIII-R4A, $p = 0.025$) and their activation was substantially shifted to more positive potentials (**P**: both $p = <0.0001$). Mean ± SEM; mutants and wt controls compared by two-tailed student's *t*-tests or one-way ANOVA followed by Dunnett's multiple comparisons test *$\triangleq < 0.05$,**$\triangleq p < 0.01$,***$\triangleq p < 0.001$. *n* number of recorded cells, *N* number of transfections.

chose to replace both residues, resulting in the construct Ca$_V$1.1_K1245A,R1249G, which, for simplicity, will be referred to as Ca$_V$1.1_VSDIV-K4A. In the background of the embryonic variant, the Ca$_V$1.1e_VSDIV-K4A mutation halved the current density but otherwise did not change the gating properties (Fig. 2I–L). In the background of the adult variant Ca$_V$1.1a_VSDIV-K4A caused a 6 mV shift of the voltage-dependence of current activation to less polarized potentials. Importantly, in neither case did mutation of the innermost gating charges alter the size or voltage-dependence of EC-coupling (Fig. 2M–P). As shown previously, the exclusion of the alternatively spliced exon 29 in the embryonic variant resulted in a substantially 20 mV left-shifted voltage dependence of current activation without a concomitant shift of EC-coupling. These experiments demonstrate that neither neutralization of the innermost gating charges, nor the insertion of exon 29 in VSD IV disturbs EC-coupling. Together these results confirm that VSDs I and IV contribute to the regulation of Ca$_V$1.1 channel gating properties, but are not critically involved in activating EC-coupling. In addition, these molecular modifications provide experimental paradigms for revealing the specific functions of individual VSDs.

## Mutating VSD III impedes EC-coupling

Therefore, we applied the same strategy to examine the importance of VSDs II and III in current gating and activation of EC-coupling. We reasoned that, alike in VSD I, neutralizing the innermost gating charges of VSDs II and III might specifically perturb the actions of these VSDs and alter the activation properties of those Ca$_V$1.1 functions regulated by the mutated VSD, i.e. calcium currents and/or EC-coupling. However, the number and nature of the gating charges in VSD II and III differ from that in VSD I (Fig. 3B, J). In VSD II the gating charge corresponding to R174 (R4) in the first domain is K537 (K4) and this VSD contains an additional gating charge (K540) in the R5 position. In VSD III the R4 gating charge is R906, in the R5 position VSD III contains an asparagine (N909) and adjacent to it an arginine in the R5 + 1 position. To exclude potential interferences of any of these residues when neutralizing R4, we chose to replace these residues with the corresponding amino acids found in VSD I. The resulting constructs are Ca$_V$1.1_K537A,K540S and Ca$_V$1.1_R906A,N909S,R910G, which, for clarity, from here on will be referred to as Ca$_V$1.1_VSDII-K4A and Ca$_V$1.1_VSDIII-R4A, respectively. To harness the advantages of the respective channel variants (superior analyses of current properties in

$Ca_V1.1e$ and minimal contamination of calcium transients by calcium influx in $Ca_V1.1a$), we analyzed the effects of the gating charge mutations in both splice variants.

Reconstitution of dysgenic myotubes with $Ca_V1.1x\_VSDII-K4A$, carrying the charge-neutralizing mutations in the second VSD, restored calcium currents and EC-coupling in both splice variants (Fig. 3A–H, Supplementary Table 1). Compared to matched wildtype controls $Ca_V1.1a\_VSDII-K4A$ displayed slowed activation kinetics and with $Ca_V1.1e\_VSDII-K4A$ the voltage-dependence of current activation was shifted to less depolarized potentials by about 12 mV. The amplitudes of depolarization-induced calcium transients ($\Delta F/F$) were similar in all wildtype and mutant constructs (Fig. 3E, G). In $Ca_V1.1e\_VSDII-K4A$, but not in $Ca_V1.1a\_VSDII-K4A$, the activation of calcium transients showed a 12 mV left-shift of $V_{1/2}$ (Fig. 3F, H). This paralleled the left shift in current activation in this mutant and likely resulted from the known contribution of the calcium influx in this channel variant[25,26]. Together, these results indicate a moderate contribution of VSD II in regulating the gating properties of $Ca_V1.1$ currents, but no direct role in the activation of EC-coupling.

In contrast, the corresponding mutations in VSD III severely affected calcium currents and EC-coupling (Fig. 3I–P, Supplementary Table 1). Expression of $Ca_V1.1a\_VSDIII-R4A$ failed to restore calcium currents and expression of $Ca_V1.1e\_VSDIII-R4A$ shifted $V_{1/2}$ of current activation to more depolarized potentials by about 18 mV (Fig. 3K, L). This is highly reminiscent of the effects on current activation observed by the corresponding mutation in VSD I (cf. Fig. 2). However, contrary to the results in VSD I, neutralizing the innermost gating charges in VSD III in both channel variants reduced the amplitudes of the calcium transients and shifted their activation substantially to more positive potentials (Fig. 3M–P). Curiously, with $Ca_V1.1e\_VSDIII-R4A$ activation of the calcium transients displayed a bi-modal voltage-dependence with $V_{1/2}$ shifts relative to wildtype of about +23 mV and +100 mV, respectively (Fig. 3N, P). Normal co-clustering of $Ca_V1.1a\_VSDIII-R4A$ and $Ca_V1.1e\_VSDIII-R4A$ with RyR1 demonstrate that these functional defects do not result from decreased expression or triad targeting of the mutated channels (Supplementary Fig. 1A, B). To our knowledge, this is the first experimental evidence showing that the perturbation of a specific VSD in $Ca_V1.1$ considerably altered the properties of depolarization-induced calcium transients and this implicates VSD III in the control of skeletal muscle EC-coupling.

## A VSD III chimera sensitizing EC-coupling

The evidence presented so far suggests that specifically, VSD III controls EC-coupling and that all four VSDs contribute to regulation of $Ca_V1.1$ channel gating. However, in VSDs II and IV the relatively weak effects on current properties of the gating charge neutralizing mutations may raise doubts about the efficacy of these mutations in perturbing the voltage-sensor action. Although in VSD IV, inclusion of the alternatively spliced exon 29, which substantially right-shifted the voltage-dependence of current activation did not affect EC-coupling (Fig. 2I–P), as reported previously[16,17]. Moreover, ectopic insertion of the IVS3-S4 loop in VSD I abolishes the specific action of that VSD, resulting in an acceleration of current kinetics[6]. Therefore, we hypothesized that ectopic insertion of the IVS3-S4 loop into VSD II and VSD III would similarly perturb the actions of these VSDs and thus reveal their specific contributions to the regulation of calcium currents and EC-coupling. Because the already right-shifted voltage dependence of the adult splice variant might mask the effects of ectopic insertion of the IVS3-S4 linker in VSDs II or III, initially we chose to perform these experiments in the background of the embryonic splice variant, generating $Ca_V1.1e\_VSDII-L_{IV}$ and $Ca_V1.1e\_VSDIII-L_{IV}$ (Fig. 4B, J). When expressed in dysgenic myotubes, both constructs restored calcium currents as well as EC-coupling (Fig. 4). Expression of $Ca_V1.1e\_VSDII-L_{IV}$ did not alter current kinetics or amplitude but resulted in an 8 mV left

shift of the voltage-dependence of current activation (Fig. 4C, D; Supplementary Table 1). This was mirrored by a similar left shift in the activation of depolarization-induced calcium transients compared to wildtype control (Fig. 4F, H), indicating that the left-shifted onset of the calcium transient in $Ca_V1.1e\_VSDII-L_{IV}$ probably is caused by the left-shifted currents, and not by direct effects on the skeletal muscle EC-coupling mechanism.

In contrast, insertion of the IVS3-S4 loop into VSD III did not alter the current properties, but the calcium transients of myotubes expressing $Ca_V1.1e\_VSDIII-L_{IV}$ showed a strikingly left-shifted and bimodal voltage-dependence of activation (Fig. 4I–P). An initial, partial activation of calcium transients occurred at about −62 mV and a second rise occurred at −14 mV (Supplementary Table 1). As the secondary rise of calcium transients coincided with the onset of current activation, this component of the calcium transient probably arises secondarily from the calcium influx through $Ca_V1.1e$ by calcium-induced calcium release.

To exclude the effects of calcium influx on cytoplasmic calcium signals, we re-examined the consequences of ectopic insertion of the IVS3-S4 loop in VSD II and VSD III on EC-coupling in the background of a non-conducting $Ca_V1.1e$ channel. The N617D mutation in the pore loop II of $Ca_V1.1$ previously has been shown to abolish calcium conduction by increasing the affinity of the selectivity filter for calcium[27]. Consequently, depolarization-induced calcium transients observed in myotubes expressing this non-conducting $Ca_V1.1nc$, per definition arise from skeletal muscle EC-coupling. Therefore, we introduced this point mutation in our two loop chimeras, generating the constructs, $Ca_V1.1nc\_VSDII-L_{IV}$ and $Ca_V1.1nc\_VSDIII-L_{IV}$. Co-clustering of $Ca_V1.1nc\_VSDIII-L_{IV}$ with RyR1 confirmed normal expression and triad targeting of the mutant construct (Supplementary Fig. 1C). As expected, expression of these constructs in dysgenic myotubes reconstituted EC-coupling, but failed to produce calcium currents. Importantly, myotubes transfected with $Ca_V1.1nc\_VSDII-L_{IV}$ activated calcium transient at the same voltages as $Ca_V1.1nc$ controls (Fig. 4A–H, Supplementary Table 1). This confirms that the small left-shift of $V_{1/2}$ observed in the conducting channel ($Ca_V1.1e\_VSDIII-L_{IV}$; see above) resulted secondarily from the equally left-shifted calcium influx and that independent of its currents VSD II is not involved in regulation of EC-coupling.

In contrast, myotubes transfected with $Ca_V1.1nc\_VSDIII-L_{IV}$ had calcium transients with comparable amplitudes as the $Ca_V1.1nc$ controls, but with the voltage-dependence of activation left-shifted by 62 mV (Fig. 4M–P, Supplementary Table 1). As opposed to the conducting version, the calcium transients activated by the non-conducting $Ca_V1.1nc\_VSDIII-L_{IV}$ displayed a monophasic voltage-dependence of activation (Fig. 4N). Apparently, the secondary rise of the calcium transient commencing at −20 mV, observed in the conducting version of this construct, resulted from the calcium currents. The striking left-shift of the voltage-dependence of the calcium transients caused by the ectopic insertion of the IVS3-S4 loop in VSD III impressively demonstrates the importance of this VSD in skeletal muscle EC-coupling. Together with the right-shift observed upon neutralization of the innermost gating charges, and the absence of effects on calcium transients with the corresponding mutations in VSD II, these results provide compelling evidence that VSD III of $Ca_V1.1$ exclusively controls skeletal muscle EC-coupling.

## Non-canonical state transitions of VSD III

Collectively, the published and current data indicate that the four VSDs of $Ca_V1.1$ respond to depolarization with distinct voltage-dependences and kinetics. All four VSDs contribute differentially to channel gating, but only the action of a single voltage sensor, VSD III, regulates depolarization-induced calcium release from the SR. Our experiments further suggest that strategically placed mutations or insertions specifically alter the state transitions of the modified VSD, consequently

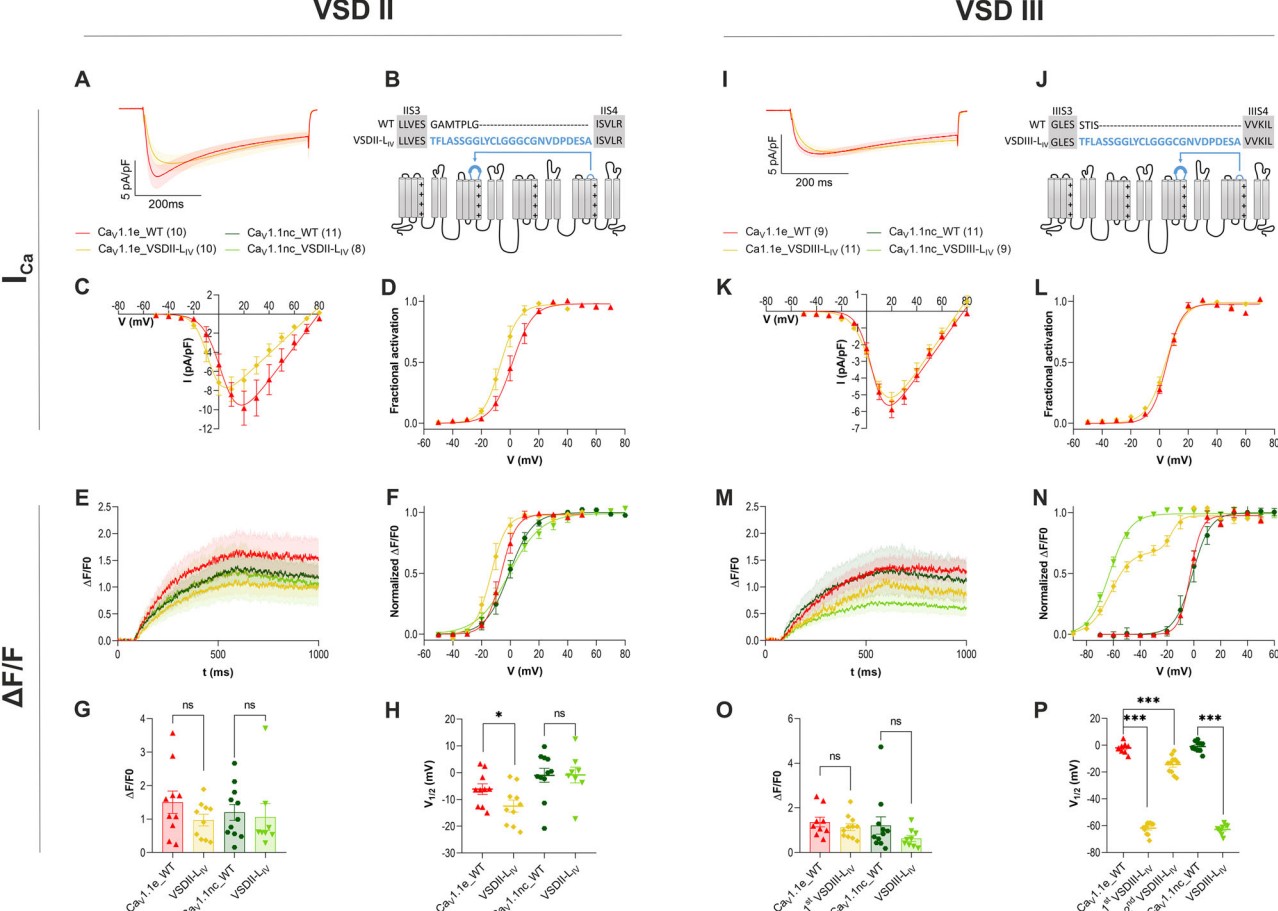

**Fig. 4 | Ectopic insertion of the IVS3-S4 linker in VSD II and VSD III of Ca$_V$1.1 differentially affects the properties of calcium currents and of EC-coupling.** In VSDs II and III, the extracellular loops connecting S3 and S4 were replaced by the corresponding sequence from VSD IV (**B**, **J**) in Ca$_V$1.1e and its non-conducting version Ca$_V$1.1nc. The chimeras were expressed in dysgenic myotubes, and calcium currents and/or depolarization-induced calcium transients were recorded. **A–H** In VSD II, insertion of the IVS3-S4 linker in Ca$_V$1.1e_VSDII-L$_{IV}$ (yellow, $n = 10/N = 7$) caused a 8 mV left-shift of the voltage-dependence of activation of the calcium currents (**D**) and of the calcium transients (**F**, **H**) relative to WT controls (red, $n = 10/N = 7$). The same insertion in the non-conducting Ca$_V$1.1nc_VSDII-L$_{IV}$ (light green, $n = 8/N = 5$) did not shift the voltage-dependence nor affected the amplitude of calcium transients, relative to matched Ca$_V$1.1nc controls (dark green, $n = 11/N = 9$) (**G**: Ca$_V$1.1e WT vs. VSDII-L$_{IV}$, $p = 0.17$; Ca$_V$1.1nc WT vs. VSDII-L$_{IV}$ $p = 0.76$; (**H**) Ca$_V$1.1e WT vs. VSDII-L$_{IV}$,

$p = 0.049$; Ca$_V$1.1nc WT vs. VSDII-L$_{IV}$, $p = 0.97$). **I–L** In VSD III, insertion of the IVS3-S4 linker in Ca$_V$1.1e-VSDIII-L$_{IV}$ did not alter the voltage-dependence of current activation. **M–P** In contrast, the voltage-dependence of depolarization-induced calcium transients was shifted by 60 mV to more negative potentials in the conducting (Ca$_V$1.1e_VSDIII-L$_{IV}$, yellow, $n = 11/N = 6$) and by 62 mV in the non-conducting (Ca$_V$1.1nc_VSDIII-L$_{IV}$, light green, $n = 9/N = 4$) chimera, compared to matched controls (Ca$_V$1.1e_WT, red, $n = 9/N = 7$; Ca$_V$1.1nc_WT, dark green, $n = 11/N = 6$), (P: both $p = <0.0001$) with little effect on the amplitude of the calcium transients (O: Ca$_V$1.1e WT vs. VSDIII-L$_{IV}$, $p = 0.4$; Ca$_V$1.1nc WT vs. VSDIII-L$_{IV}$, $p = 0.2$). Mean ± SEM; mutants and wt controls compared by two tailed student's $t$-tests or one-way ANOVA followed by Dunnett's multiple comparisons test *≙$p < 0.05$,**≙$p < 0.01$,***≙$p < 0.001$. $n$ number of recorded cells, $N$ number of transfections.

altering its specific contribution to channel gating and/or EC-coupling. To examine these predictions of our experiments on the molecular level, we turned to MD simulations. Ca$_V$1.1 was the first member of the eukaryotic Ca$_V$/Na$_V$ family for which the structure has been solved at atomic resolution[4,28]. Its cryo-EM structure shows the VSDs in the activated (up-) state. At present, no resting state structures of eukaryotic Ca$_V$ or Na$_V$ channels are available and the trajectories and kinetics of their state-transitions remained obscure. Therefore, we utilized MD simulations of Ca$_V$1.1 packed in a lipid environment exposed to an electric field to model the voltage-dependent movements of its VSDs out of the known activated state into resting states. As predicted by the sliding helix model, within 2 μs simulation time all four S4 helices move downward relative to the positions of the other transmembrane helices (Fig. 5A). During this transition up to two of its gating charges (R3 and R2) passed the conserved phenylalanine, which marks the hydrophobic constriction site (HCS) (Fig. 5B). Other features known from prokaryotic channel structures, like the formation of 3$_{10}$ helix segments and transient ion-pair formation between gating charges and

countercharges in the adjacent helices[2], were recapitulated in our MD simulations. However, in the pseudo-heterotetrameric Ca$_V$1.1 the trajectories and speeds at which the S4 helices moved downward, showed striking differences between the four VSDs (Fig. 5C). Consistent with its known role in determining the slow activation kinetics of Ca$_V$1.1[15], VSD I was the slowest VSD reaching a potential resting state only after 2 μs. VSD III, on the other hand, displayed the fastest S4 movement of all four VSDs. It precipitously dropped down into the final resting state within less than 0.5 μs. This rapid response to changes in the membrane potential agrees with experimental evidence from a voltage-clamp fluorometry study of Ca$_V$1.1 in oocytes[7] and is consistent with the rapid onset of skeletal muscle EC-coupling. These distinct characteristics of the S4 kinetics could be reproduced in multiple runs of MD simulations. Remarkably, the simulated resting state structures of Ca$_V$1.1 VSDIII closely resembles the cryo-EM structure of a mutated Na$_V$Ab/Nav1.7 channel chimera captured in the resting state by disulfide locking (Cα RMSD of 1.2 Å)[29]; and our intermediate state structure 2 resembles the cryo-EM structure of a Na$_V$Ab/Nav1.7 channel

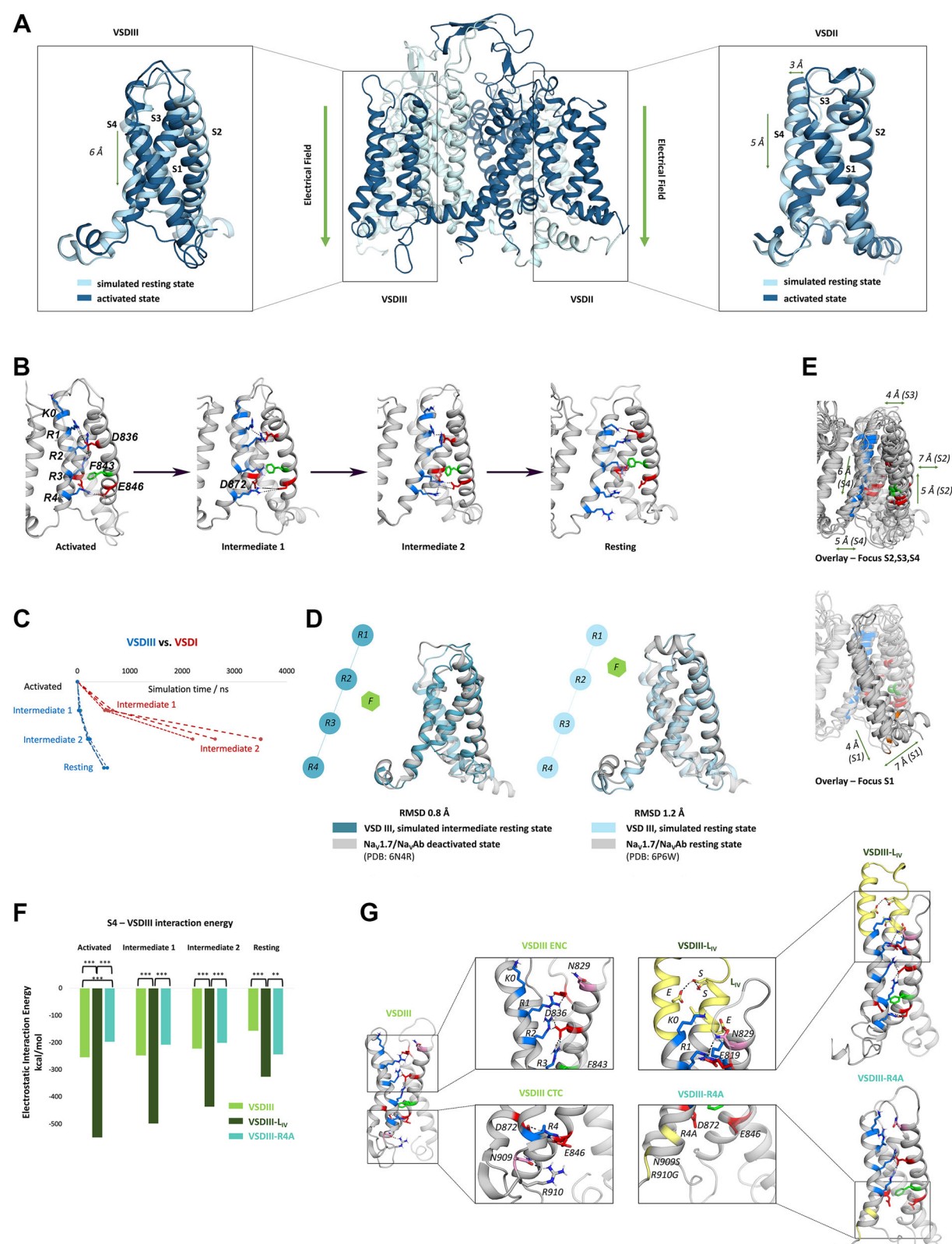

chimera stabilized in a deactivated state by toxin binding (Cα RMSD of 0.8 Å)[30] (Fig. 5D). While the state-transitions of all four VSDs displayed many features characteristic of the sliding helix model, we noted unique structural features of the VSD III movement, distinguishing it from the other three VSDs. In these VSDs the S1, S2 and S3 helices function as essentially stationary pivots along which the S4 helix moves up and down. In contrast, in VSD III of Ca$_V$1.1 all four

transmembrane helices undergo considerable vertical and lateral rearrangements; including a 4 Å downward and 7 Å sideways swinging of the cytoplasmic end of S1 (Fig. 5E, Supplementary Table 2). This suggests that VSD III specifically employs a non-canonical conformational voltage-sensing mechanism combined with the sliding helix motion. Thus, both its rapid response to voltage changes and its complete structural rearrangement set VSD III apart from the other

**Fig. 5 | MD simulation of state transitions of Ca$_V$1.1 VSDs in response to an electric field. A** Side view of Ca$_V$1.1a and structure alignments of VSD III (left) and VSD II (right) before and after equilibration in an electric field for 2.5 µs. **B** Activated-, two intermediate-, and resting states of Ca$_V$1.1 VSD III displaying the downward movement of S4 relative to F843 (green) and the transient interactions formed by its positive gating charges (K0-R4, blue) with negative countercharges (red) in the neighboring helices (*n* = 3). **C** Simulation times required by VSD I (red) and VSD III (blue) to reach their respective intermediate and resting states. **D** Overlay of the intermediate state 2 and the resting state structures with two experimentally determined resting state structures of Na$_V$Ab/Na$_V$1.7 chimeras (PDB: 6N4R, 6W6D). **E** Overlay of the four states of VSD III displaying considerable rearrangements of all four helices during the voltage sensing process. Upper, highlighting S2, S3, S4; lower highlighting S1. **F, G** Comparison of interaction energies (**F**) and molecular interactions (**G**) stabilizing the activated state of wildtype VSD III (left), Ca$_V$1.1e_VSDIII-L$_{IV}$ (upper right) and Ca$_V$1.1a_VSDIII-R4A (lower right, *n* = 3) (**F**: for all comparisons *p* = <0.0001). Mutants and WT controls were compared by two-tailed *t*-tests or one-way ANOVA *≙ *p* <0.05,**≙ *p* < 0.01,***≙ *p* < 0.001.

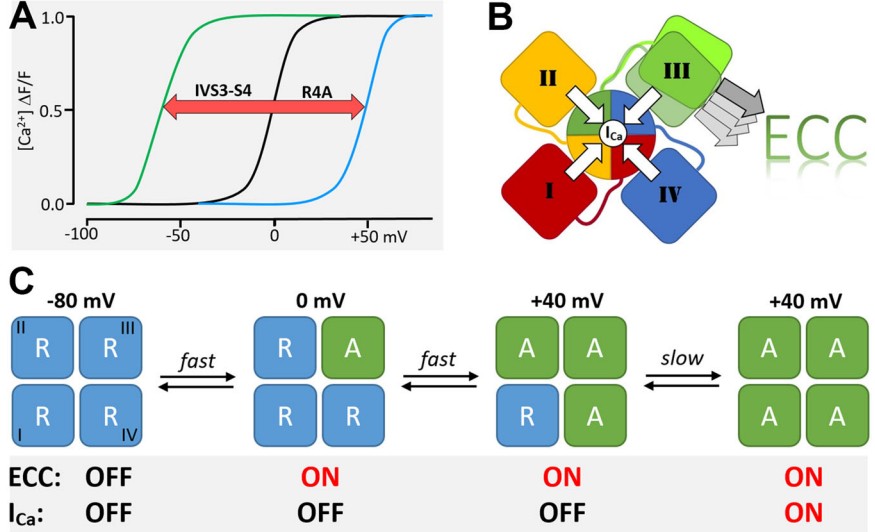

**Fig. 6 | Actions of Ca$_V$1.1 VSDs in activating EC-coupling and channel gating. A** Mutations which exclusively alter the properties of VSD III can tune the voltage-dependence of EC-coupling in a range of >100 mV. **B** All four VSDs contribute to channel gating, but only the non-canonical voltage-sensing mechanism of VSD III activates EC-coupling. **C** Reaction scheme of Ca$_V$1.1 VSDs (R, resting-, A, activated state): Depolarization to 0 mV rapidly activates VSD III, which is sufficient for activating EC-coupling. Further depolarization to +40 mV rapidly activates VSD IV, but only after the slow VSD I reaches its activated state the channel opens. VSD II may activate together with VSD III or IV and is necessary for channel gating but not for EC-coupling.

VSDs and make it particularly suited for initiating the conformational coupling mechanism of skeletal muscle EC-coupling.

If these kinetic and structural characteristics of the VSD movements revealed by the MD simulations accurately reflect the specific properties of the individual VSDs of Ca$_V$1.1 during voltage-dependent state transitions, we also expect our simulations to display the altered activation properties observed experimentally in the mutated channels. The most striking effect observed experimentally, was the dramatic left shift of the voltage-dependence of activation of the calcium transients seen with Ca$_V$1.1e_VSDIII-L$_{IV}$ and Ca$_V$1.1nc_VSDIII-L$_{IV}$. Therefore, we introduced this mutation in our structure models and examined the state transitions of the VSDs in the electric field. Indeed, MD simulations of Ca$_V$1.1e with the IVS3-S4 loop sequence inserted in VSD III (resembling Ca$_V$1.1e_VSDIII-L$_{IV}$) displayed several additional interactions between the extended S4 helix and the neighboring S2 and S3 helices that substantially increase the interaction energies and stabilize the activated state (Fig. 5F, G). In contrast, the charge-neutralizing mutation of the innermost gating charges in VSD III, which caused a specific right shift of the voltage-dependence of EC-coupling, displayed reduced stabilizing interactions in the charge transfer center. This was accompanied by a reduction of interaction energies in the activated state and an increase in the resting state. Together these MD simulations of the mutated channel constructs indicate that the measured left- and right-shifts in the voltage-dependence of activation of EC-coupling are brought about by the increased and decreased stabilization, respectively, of VSD III in the activated state. Notably, the state transitions of the other VSDs were not affected by these mutations, indicating that the functional effects on EC-coupling are directly caused by the altered movement of VSD III and do not require allosteric interactions with other VSDs.

## Discussion

Together the mutagenesis experiments and structure modeling identify VSD III of Ca$_V$1.1 as the sole voltage sensor controlling skeletal muscle EC-coupling (Fig. 6A). Mutations at the cytoplasmic end of IIIS4, which destabilize its activated state, right-shift the voltage-dependence of depolarization-induced calcium transients by about 45 mV. The ectopic insertion of the IVS3-S4 loop sequence at the extracellular end of IIIS4 left-shifts the voltage-dependence of calcium release by more than 60 mV, presumably by stabilizing the activated state. The opposite directions of the observed voltage shifts can be explained by the differential effects of the two mutations on VSD activation. Our structure models show that the inner gating charges of VSD III (R4, R5 + 1) form multiple ionic interactions with the countercharges of the charge transfer center in intermediate state 1 and the activated state. Neutralizing R4 and R5 + 1 abolishes these interactions, thus destabilizing the activated state and right-shifting the voltage-dependence of EC-coupling. On the contrary, insertion of the IVS3-S4 loop creates additional interactions between the extended S3 and S4 helices, which stabilize the activated state and cause the observed left-shifted voltage-dependence of EC-coupling. Notably, this left-shift was not recapitulated in the voltage-dependence of current activation, indicating that VSD III is not limiting for this current property, but

that voltage-dependence of current activation is limited by one or more other VSDs.

Conversely, the corresponding mutations in the other VSDs exclusively affect current properties but not calcium transients[6,16], indicating that VSDs I, II and IV do not participate in the regulation of EC-coupling. MD simulations of the state transitions of $Ca_V1.1$ in response to an electric field indicate that VSD III is the fastest of the four VSDs. This is consistent with direct measurements of $Ca_V1.1$ S4 kinetics in voltage-clamp fluorometry recordings of $Ca_V1.1$ in oocytes, showing that only the kinetics and voltage-dependence of VSD III match the properties of EC-coupling, while the other VSDs either respond too slowly or at more depolarized membrane potentials than EC-coupling[7]. Unexpectedly, our MD simulations indicate that, when exposed to an electric field, VSD III undergoes a non-canonical movement involving considerable displacements of all four transmembrane helices. The cytoplasmic loop connecting the pore domain of repeat II with VSD III contains the sequence critical for skeletal muscle EC-coupling and for functional interaction with RyR1 and STAC3[31–33]. Now between the resting and activated state the cytoplasmic end of IIIS1 performs a 4 Å downward and 7 Å sideways swinging motion. Thus, it is plausible to assume that this motion of IIIS1 actuates the conformational coupling of the $Ca_V1.1$ II-III loop to the RyR1, just as the motion of the S4 helices actuates the gating of the channel pore (Fig. 6B). While the canonical sliding helix motion with essentially stationary S1, S2 and S3 helices would hardly affect the conformation of the II-III loop, the unique global rearrangement of VSD III might represent a necessary requirement of skeletal muscle EC-coupling.

Contrary to the activation of EC-coupling solely by VSD III, all four VSDs appear to be involved in current gating (Fig. 6C). Previously, we and others demonstrated the importance of VSDs I and IV for determining the specific $Ca_V1.1$ current properties[6,14–18,20,34]. Our current results add that also VSDs II and III contribute to channel gating. In VSD II neutralization of the inner gating charges and insertion of the IVS3-S4 loop resulted in modest changes of kinetics and voltage-dependence of activation, respectively. Apparently, VSD II participates in the regulation of current gating properties, but its contribution is comparatively small. In VSD III, the right shifts of the voltage-dependence of calcium transients observed in $Ca_V1.1a$_VSDIII-R4A and $Ca_V1.1e$_VSDIII-R4A were accompanied by comparable right-shifts of the calcium currents, indicating the importance of VSD III for current gating. On the other hand, the dramatic left-shift of the voltage-dependence of the calcium transients observed in $Ca_V1.1e$_VSDIII-$L_{IV}$ was not accompanied by a change in voltage-dependence of current activation. While effects on current kinetics observed with this construct support a role of VSD III in regulating gating properties, its effect on the voltage-dependence of current activation may be masked by the limiting action of another VSD. This observation highlights a significant difference resulting from the differential operational modes of EC-coupling and channel gating by one or several VSDs, respectively. Where a single VSD controls activation and deactivation of a process, like VSD III for depolarization-induced calcium release, any change in the action of this VSD will translate into a corresponding change of its cellular function (i.e. EC-coupling). Where several VSDs jointly control a process, like all four VSDs´ roles in channel gating, the contributions of each individual VSD may differ in mode (kinetics, voltage-dependence) and magnitude. Depending on the mode of cooperation, actions of individual VSDs may be masked by limiting actions of other VSDs (obligatory model) or influence each other´s action (allosteric model). The singular effect of mutations in VSD III does not indicate any allosteric effects of other VSDs on the regulation of EC-coupling. Also, most effects on current gating properties observed in the present study, can readily be explained by obligatory cooperation of the four VSDs, which contribute differentially to the kinetics and voltage-dependence of current gating.

In conclusion, this study highlights that the homologous but structurally distinct VSDs of eukaryotic calcium channels differ from one another functionally in their specific responses to membrane potentials and, in the case of $Ca_V1.1$, in their differential contributions to its two functions as voltage-gated calcium channel and as voltage sensor of skeletal muscle EC-coupling. Moreover, our advances in structure modeling of state-transitions in response to membrane potentials yield unprecedented mechanistic insights in the molecular mechanisms underlying the voltage-sensing functions of voltage-gated ion channels. Knowing the specific VSD of $Ca_V1.1$ that controls EC-coupling, and seeing the conformational changes initiating this process, paves the way to unravel the downstream processes linking $Ca_V1.1$ VSD III movement to opening of the calcium release channel during skeletal muscle contraction.

## Methods
### Plasmids
Cloning procedures for GFP-$Ca_V1.1a$_WT and GFP-$Ca_V1.1e$_WT (protein Uniprot P07293 and cDNA Genbank NM_001101720) were previously described[16,23]. Sanger sequencing (Eurofins Genomics) verified the integrity of the newly generated plasmid.

To generate **GFP-$Ca_V1.1a$_VSDI-R4A**, the R174A mutation was introduced into GFP-$Ca_V1.1a$ by splicing by overlap extension (SOE) PCR. Briefly, the cDNA sequence of rabbit $Ca_V1.1$ (nt 1–1006) was amplified in separate PCR reactions using GFP-$Ca_V1.1a$_WT as template with overlapping primers mutating the c.520 C > G and c.521 G > C. The two separate PCR products were then used as templates for a PCR reaction with flanking primers to connect the nucleotide sequences. The resulting fragment was then SalI/EcoRI digested and ligated into the corresponding sites of GFP-$Ca_V1.1a$_WT. To generate **GFP-$Ca_V1.1e$_VSDI-R4A**, the cDNA region containing the alternatively spliced region was isolated from GFP-$Ca_V1.1e$ by XhoI/BglII digestion (nt 2653-4430) and inserted in the corresponding sites of GFP-$Ca_V1.1a$_VSDI-R4A.

**GFP-$Ca_V1.1a$_VSDII-K4A** was generated by SOE-PCR. Briefly, nt 1006–2653 of $Ca_V1.1a$ were PCR amplified with overlapping primers introducing the mutations c.1609 A > G and c.1610 A > C (to mutate K537to A) and c.1619 A > G and c.1620 G > C (to mutate K540 to S) in separate PCR reactions using GFP-$Ca_V1.1a$_WT as template. The two separate PCR products were then used as templates for a final PCR reaction with flanking primers to connect the nucleotide sequences. This fragment was then EcoRI/XhoI digested and cloned into the respective sites of GFP-$Ca_V1.1a$_WT. To generate **GFP-$Ca_V1.1e$_VSDII-K4A**, the cDNA region containing the alternatively spliced region was isolated from GFP-$Ca_V1.1e$ by XhoI/BglII digestion (nt 2653-4430) and inserted in the corresponding sites of GFP-$Ca_V1.1a$_VSDI-R4A.

To generate **GFP-$Ca_V1.1a$_VSDIII-R4A** and **GFP-$Ca_V1.1e$_VSDIII-R4A**, the R906A, N909S and R910G mutations were introduced into GFP-$Ca_V1.1a$ by SOE-PCR. The cDNA sequence of rabbit $Ca_V1.1$ (nt 2653-4430) was amplified in separate PCR reactions using GFP-$Ca_V1.1a$_WT or GFP-$Ca_V1.1e$_WT as template with overlapping primers mutating c.2716 C > G and c.2717 G > C for R906A, c.2726 A > G for N909S and c.2728 A > G for R910G. The two separate PCR products were then used as templates for a PCR reaction with flanking primers to connect the nucleotide sequences. The resulting fragment was then XhoI/BglII digested and ligated into the corresponding sites of GFP-$Ca_V1.1a$_WT.

To generate **GFP-$Ca_V1.1a$_VSDIV-R4A** and **GFP-$Ca_V1.1e$_VSDIV-R4A**, the K1245A and R1249G mutations were introduced into GFP-$Ca_V1.1a$ by SOE-PCR. The cDNA sequence of rabbit $Ca_V1.1$ (nt 2653-4430) was amplified in separate PCR reactions using GFP-$Ca_V1.1a$_WT or GFP-$Ca_V1.1e$_WT as template with overlapping primers mutating c.3733 A > G and c.3734 A > C for K1245A and c.3745 C > G for R1249G. The two separate PCR products were then used as templates for a PCR reaction with flanking primers to connect the nucleotide sequences. The resulting fragment was then XhoI/BglII

digested and ligated into the corresponding sites of GFP-Ca$_V$1.1a_WT.

To generate **GFP-Ca$_V$1.1nc_WT**, the cDNA region containing the alternatively spliced region was isolated from GFP-Ca$_V$1.1e by XhoI/BglII digestion (nt 2653-4430) and inserted in the corresponding sites of GFP-Ca$_V$1.1a_N617D[35].

To generate **GFP-Ca$_V$1.1e_VDSII-L$_{IV}$** and **GFP-Ca$_V$1.1nc_VDSII-L$_{IV}$**, the S3-S4 linker of VSDII was replaced by the cDNA sequence encoding the S3-S4 linker of VDS IV. The cDNA sequence of rabbit Ca$_V$1.1 (nt 1006−2653) was amplified in separate PCR reactions using GFP-Ca$_V$1.1e_WT or GFP-Ca$_V$1.1nc_WT as template with overlapping primers introducing the cDNA sequence encoding the S3-S4 linker of VDS IV. The two separate PCR products were then used as templates for a PCR reaction with flanking primers to connect the nucleotide sequences. The resulting fragment was then EcoRI/XhoI digested and ligated into the corresponding sites of GFP-Ca$_V$1.1e_WT.

**GFP-Ca$_V$1.1e_VDSIII-L$_{IV}$** was generated by replacing the S3-S4 linker of VSDIII with that of the VSD IV. The cDNA sequence of rabbit Ca$_V$1.1 (nt 2653-4430) was amplified in separate PCR reactions using GFP-Ca$_V$1.1e_WT as template with overlapping primers introducing the cDNA sequence encoding the S3-S4 linker of VDS IV. The two separate PCR products were then used as templates for a PCR reaction with flanking primers to connect the nucleotide sequences. The resulting fragment was then XhoI/BglII digested and ligated into the corresponding sites of GFP-Ca$_V$1.1e_WT. To generate **GFP-Ca$_V$1.1nc_VDSIII-L$_{IV}$**, the cDNA region containing the alternatively spliced region was isolated from GFP-Ca$_V$1.1e_VDSIII-L$_{IV}$ by XhoI/BglII digestion (nt 2653-4430) and inserted in the corresponding sites of GFP-Ca$_V$1.1nc_WT.

The primer sequences devised for the production of the different plasmids are listed in Supplementary Table 3.

## Cell culture and transfection

Dysgenic ($\alpha_1$s-null) myoblast (GLT) cell line[36] were plated in growth medium Dubecco's Modified Eagle Medium (DMEM) 1 g/L glucose supplemented with 10% fetal calf serum and 10% heat-inactivated horse serum (HS) in culture flasks and kept at 37 °C in a humidified incubator with 10% $CO_2$. The cells were seeded into 35 mm plastic dish for electrophysiology or gelatin and carbon-coated plastic dishes for immunofluorescence analysis. The second day and at alternated days, the medium was changed to fusion medium (DMEM supplemented with 2% HS). On the fourth day, the cells were transiently transfected with a given Ca$_V$1.1 construct using FuGeneHD transfection reagent (Promega) according to manufacturer's instructions. The cells were used for electrophysiology experiments on days 7 and 8 after plating while the immunofluorescence staining was performed on days 9 and 10 after plating.

## Immunostaining and quantification

Paraformaldehyde-fixed cultures were immunolabeled with the polyclonal rabbit anti-GFP antibody (serum, 1:10,000; A6455 Thermofisher Scientific) and the monoclonal mouse anti-RyR (34 C, 1:500; MA3-925 Thermofisher Scientific) and subsequently fluorescently labeled with Alexa-488 and Alexa-594, respectively, as previously described[37].

14-bit images were recorded with a cooled CCD camera (SPOT) and Metaview image processing software. Image composites were arranged in Adobe Photoshop CS6 and linear adjustments were performed to correct black level and contrast. Clusters of GFP-Ca$_V$1.1 and RyR were quantified using Image J software (NIH), as previously described[38]. Briefly, myotubes were selected by a ROI tool and converted to binary images using the intermodes threshold, so that only clusters are included. Using the Analyze Particle function, the numbers of particles larger than 0.198 µm$^2$ in the binary image were counted as clusters. The numbers of clusters per 100 µm$^2$ were calculated and are represented in the graphs. To quantify the GFP-Ca$_V$1.1/RyR colocalization, the thresholded image was used as a mask on the myotube image to include only the clusters and Pearson´s coefficients for colocalization were calculated by JaCoP, as previously described[39]. For each condition, the number of clusters of at least 20 myotubes from two separate experiments was counted. Graphs and statistical analysis (Student's T test) were performed using Graphpad software.

## Electrophysiology

Calcium currents were recorded with the ruptured whole-cell patch-clamp technique in voltage-clamp mode. The patch pipettes (borosilicate glass, Harvard Apparatus, Holliston, MA) had resistance of 1.5−3.5 MΩ when filled with (mM) 145 Cs-aspartate, 2 MgCl2, 10 HEPES, 0.1 Cs-EGTA, 2 Mg-ATP, and 0.2 Clabryte-590 potassium salt to record calcium transients (pH 7.4 with CsOH). The extracellular bath solution contained (mM) 10 CaCl2, 145 tetraethylammonium chloride, 10 HEPES (pH 7.4 with CsOH).

The recordings reported in Fig. 2 were performed with HEKA EPC 10 USB amplifier (Harvard Bioscience Inc.). Data acquisition and command potentials were controlled by PATCHMASTER NEXT (version 1.2, Harvard Bioscience Inc.). The recordings reported in Figs. 3 and 4 were acquired with Axopatch 200 A amplifier (Axon Instruments, Foster City, CA). Data acquisition and command potentials were controlled by pClamp software (version 10.7, Axon Instruments). The voltage-step protocol is preceded by a 1-second long pre-pulse to −30 mV in order to inactivate potential T-types conductance expressed in the myotubes sarcolemma. Subsequently, the cells are clamped at a holding potential (V$_{hold}$) of −80 mV, followed by the command potential (V$_{cmd}$) of 500 ms, ranging from −70 mV to +80 mV with an increment of 10 mV. In order to avoid calcium transient stimulation artifacts of the left-shifted constructs, the pre-pulse was omitted and the protocol was adjusted to accommodate the shift of the fluorescent calcium signal to lower voltages.

The current-voltage dependence was calculated according to:

$$I = G_{max}*(V - V_{rev})/(1 + exp(-(V - V_{1/2})/K)) \qquad (1)$$

Where $G_{max}$ is the maximum conductance of the L-type calcium channels, $V_{rev}$ is the extrapolated reversal potential of the current, $V_{1/2}$ is the potential for half-maximal conductance, and $k$ is the slope.

The conductance was extrapolated using:

$$G = (-I*1000)/(V_{rev} - V) \qquad (2)$$

The conductance voltage dependence was calculated according to the Boltzmann distribution:

$$G = G_{max}/(1 + exp(-(V - V_{1/2})/K)) \qquad (3)$$

The mean ± SEM for the calcium currents (I$_{Ca}$) sample traces was calculated by selecting the sweep with the V$_{max}$ for each recording constituting the dataset.

## Calcium recording

Simultaneously to electrical stimulation, the myoplasmic calcium concentration was recorded in reconstituted GLT myotubes utilizing a fluorescence calcium indicator (Calbryte 590 potassium salt; AAT Bioquest) included to the intracellular solution, at a concentration of 0.2 mM. The fluorescence was excited at 580 nm and the emission recorded at 593 nm using the micro-photometer photomultiplier tube PMT (IonOptix; Fig. 2) and PTI D-104 (HORIBA scientific; Figs. 3, 4). Fluorescent data acquisition was controlled by PATCHMASTER NEXT (version 1.2, Harvard Bioscience Inc.) and FelixGX (version 4.3.6904; HORIBA scientific) respectively. The fluorescent calcium signal was recorded during the holding potential (V$_{hold}$) and the command potential (V$_{cmd}$). For each protocol step, F0 was calculated as the mean of the fluorescent intensity at V$_{hold}$. ΔF was determined as the

difference between F0 and the mean at the peak fluorescent intensity at $V_{cmd}$. ΔF/F0 was plotted against the $V_{cmd}$ yielding the relationship between the relative change of fluorescent intensity and the voltage at which the signal is recorded. From this, we can extrapolate the voltage dependence of the fluorescent calcium transient as:

$$\Delta F/F0 = (\Delta F/F0)_{max}/(1+exp(-(V-V_{1/2})/K)) \qquad (4)$$

where $(\Delta F/F0)_{max}$ is the maximum fluorescence change, $V_{1/2}$ is the potential at the half-maximal fluorescence change, and $k$ is the slope. The bimodal voltage-dependence of the calcium transients signal of $Ca_V1.1e\_VSDIII$-R4A and $Ca_V1.1e\_VSDIII$-$L_{IV}$ (Figs. 3, 4) were calculated according to:

$$\Delta F/F0 = (\Delta F/F0)_{max1}/(1+exp(-(V-V_{1/2*})/K_1))$$
$$+ ((\Delta F/F0)_{max2}/(1+exp(-(V-V_{1/2**})/K2)) \qquad (5)$$

where $(\Delta F/F0)_{max1}$ and $(\Delta F/F0)_{max2}$ are the maximum fluorescence change, $V_{1/2*}$ and $V_{1/2**}$ are the potentials at the half-maximal fluorescence change and $K_1$ and $K_2$ are the slopes of the first and of the second sigmoidal curve respectively.

The mean ± SEM for the sample traces of the fluorescent calcium signal was calculated for the sweep at $V_{max}$ for each recording. In Fig. 4M, the sample trace for $Ca_V1.1\_VSDIII$-$L_{IV}$ was calculated as mean ± SEM of the sweep clamped at −40 mV.

### Molecular modeling

We predicted the structure of the human WT $Ca_V1.1$ $α_1$ subunit by making a homology model based on the cryo-EM structure of the rabbit $Ca_V1.1$ $α_1$ subunit with the VSDs in the up-state and the pore closed[28]. Homology modeling has been performed using Rosetta and MOE (Molecular Operating Environment, version 2020, Molecular Computing Group Inc, Montreal, Canada). The sequence identity between the rabbit and the human $Ca_V1.1$ α1 subunit is 92.6%, the sequence similarity even 95.6%. Because of the high sequence similarity and identity between the human and the rabbit $Ca_V1.1$, we generated only 10 homology models and chose the one model with the best energy score as starting structure for further minimizations, equilibrations, and simulations. The fragment-based cyclic coordinate descent algorithm implemented in Rosetta was used to generate structures for loops that were not resolved in the $Ca_V1.1$ $α_1$ subunit template[40,41]. The C-terminal and N-terminal parts of each domain were capped with acetylamide and $N$-methylamide to avoid perturbations by free charged functional groups. The structure was aligned in the membrane using the PPM server[42] and inserted into a plasma membrane consisting of POPC (1-palmitoyl-2-oleoyl-sn-glycero-3-phosphocholine) and cholesterol in a 3:1 ratio, using the CHARMM-GUI Membrane Builder[43]. Water molecules and 0.15 M $CaCl_2$ were included in the simulation box. For calcium the standard parameters for calcium-ions were replaced with the multi-site calcium of Zhang et al.[44]. This multi-site model has been used to investigate calcium permeation in a number of channels, including type-1 ryanodine receptor[44–48].

### Molecular dynamics simulations

All simulations of the WT and the VSDIII-R4A and VSDIII-$L_{IV}$ variants were performed using GROMACS 2020.2[49,50] with the CHARMM36m force field for the protein, lipids and ions (except for calcium as described above)[51]. The TIP3P water model was used to model solvent molecules[52]. The system was minimized and equilibrated using the suggested equilibration input scripts from CHARMM-GUI[53], i.e., the system was equilibrated using the NPT ensemble for a total time of 2 ns with force constraints on the system components being gradually released over six equilibration steps. The systems were further

equilibrated by performing a 10 ns simulation with no electric field applied. The temperature was maintained at T = 310 K using the Nosé-Hoover thermostat with the inverse friction constant set to 1.0 ps[54], and the pressure was maintained semi-isotropically at 1 bar using the Parrinello-Rahman barostat with the period of pressure fluctuations at equilibrium set to 5.0 ps and the compressibility set to 4.5e-5 bar$^{-1}$ [55]. Periodic boundary conditions were used throughout the simulations. Long-range electrostatic interactions were modeled using the particle-mesh Ewald method[56] with a cut-off of 12 Å. The LINCS algorithm[57] was used to constrain bond lengths involving bonds with hydrogen atoms. The simulations were performed using a time step of 2 fs and an applied electric field to produce membrane voltage[58]. Fields of 600 mV were applied respectively. Such unphysiologically high potentials are required to observe the VSD state transitions, which usually occur at the ms time scale, within the few μs simulation time. This did not compromise the overall integrity of the VSD fold and the high structural similarity to existing resting-state structures of related channels further validated the structures resulting from our MD simulation (see Fig. 5D). All applied field simulations were ~2.5 μs long and repeated three times. PyMOL Molecular Graphics System was used to visualize the key interactions and point out differences in the WT and mutant structures (PyMOL Molecular Graphics System, version 2.0, Schrödinger, LLC).

### Statistics

SigmaPlot (version 12.0; SPSS) was used for statistical analyses and curve fitting; GraphPad Prism (version 8.0.1; Graphpad Softaware LLC) and CorelDRAW2021 (version 23.0.0.363; Corel Corporation) were used to make the figures. All data are presented as mean ± SEM. Statistical comparison of the fit parameters were obtained by using either Student's $t$ test or one-way ANOVA combined with Dunnett's Multiple comparison post hoc test with significance criteria $*p < 0.05$, $**p < 0.01$, $***p < 0.001$, $****p < 0.0001$.

### Reporting summary

Further information on research design is available in the Nature Portfolio Reporting Summary linked to this article.

## Data availability

The data generated or analyzed during this study are included in this published article. The structures used in this manuscript are publicly available, with the PDB codes: 5GJW, 6P6W, 6N4R respectively. Source data are provided with this paper.

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

## Acknowledgements

We thank Nicole Kranebitter, Maria Kharitonova, Sabrina Pennelli and Georg Kern for technical help. This research was funded in part by the Austrian Science Fund (FWF) Grant-DOI 10.55776/P35618 to B.E.F, and Grant-DOI 10.55776/P33776 to M.C., and by the Austrian Academy of Sciences APART-MINT postdoctoral fellowship to M.L.F.Q. For open access purposes, the authors have applied a CC BY public copyright license to any author accepted manuscript version arising from this submission. The computational results presented have been achieved [in part] using the Vienna Scientific Cluster (VSC). S.P. and M.C.H. are students of the $Ca_VX$ PhD program co-funded by FWF (DOC30) and the Medical University Innsbruck.

## Author contributions

Experiments were conceived and designed by S.P., M.C., and B.E.F. Experiments: Electrophysiology and calcium recording were performed by S.P. and M.C.H. Structure modeling was performed by M.L.F.Q. Generation of expression plasmids and immunofluorescence analyses were planned and supervised by M.C. Data were analyzed and figures prepared by S.P., M.C.H., M.L.F.Q., M.C., and B.E.F. Cow-orkers were trained and the project was supervised by B.E.F., M.C., Y.E.G., P.T., and K.R.L. The paper was written by B.E.F. with input from all authors.

## Competing interests

The authors declare no competing interests.
