## [Peer Review File · Nature Communications]

CaV1.1 voltage-sensing domain III exclusively controls skeletal muscle excitation-contraction couplingREVIEWER COMMENTS

Reviewer #1 (Remarks to the Author):

Pelizzari et al describe experiments that evaluate the relative role of the four voltage sensor domains of CaV1.1 (VSD I, VSD II, VSD III, and VSD IV) on the calcium channel (pore opening, calcium current) and voltage sensor (triggering SR calcium release) functions of CaV1.1.

The authors measured voltage-dependent calcium currents and calcium transients following transient expression of CaV1.1 with neutralizing mutations (e.g. R -> A) to the inner most charged residues within the S4 domains of VSDs I-IV in dysgenic (CaV1.1-null) myotubes. Immunocytochemical studies confirmed proper junctional targeting and co-localization with the ryanodine receptor. The functional studies indicate that while all four VSDs differentially impact CaV1.1 calcium current properties (e.g. kinetics, magnitude, voltage dependence), the magnitude and voltage dependence of CaV1.1-mediated SR calcium release are controlled only by VSD III. Molecular dynamics studies of CaV1.1 within a lipid environment and a focused electric field found that the S4 segment of each VSD exhibited unique trajectories and speed during changes in membrane voltage (e.g. VSD I was the slowest and VSD III was the fastest). Hyperpolarization and depolarization shifts in the voltage-dependence of calcium release resulting from the mutations introduced in VSD III could be explained by increased and decreased stabilization, respectively, of VSD III in the activating state. Most intriguingly, VSD III movement was unique among the four VSDs in exhibiting both vertical and non-canonical lateral transitions during activation, which presumably promotes conformational coupling of the adjacent II-III loop and subsequent opening of the ryanodine receptor during EC coupling.

These results extend the authors' prior findings regarding the impact of VSDs I and IV on CaV1.1 channel function (PMID: 26749449, 27185857) and are consistent with results reported by others in showing the rapid movement of VSD III at negative voltages (PMID: 34546289) and the critical role of C-terminal end of the CaV1.1 II-III loop in EC coupling (PMID: 9737952, 14627713). The manuscript could be improved by the authors addressing the following points.

1. Surprisingly, no charge movement data were included in this study. As the S4 (and S3-S4 linker) mutations used in this study are predicted to exhibit specific effects on a given VSD (e.g. CaV1.1e VSDIII-LIV mutant is assumed to specifically alter VSD III function), effects on charge movement magnitude and voltage dependence could be linked to the behavior of a specific VSD. It would be most informative to assess charge movements for non-conducting versions of VSD III mutants that result in >100 mV shift in the voltage dependence of calcium release (e.g. comparing CaV1.1nc_VSDIII-R4A with CaV1.1nc_VSDIII-LIV). Assuming the specificity assumption above applies, differences in the voltage-dependence of charge movement for these two constructs would reflect changes in the voltage-dependence of VSD III, which could then be correlated with the simultaneously recorded calcium transients. For example, how closely does the voltage dependence of CaV1.1 VSDIII charge movement reflect the voltage dependence (V_{0.5} and k) of calcium release.
2. The authors do not discuss reasons for why the maximum magnitude of calcium release (ECC) is reduced by >90% for the two VSDIII-LIV constructs shown in Fig. 4O (CaV1.1e_VSDIII-LIV and CaV1.1nc_VSDIII-LIV)? This is one of the largest effects observed in this study, and surprisingly, is essentially ignored by the authors. While ICC results confirm normal junctional targeting of these constructs, could inclusion of LIV limit tetrad formation or otherwise largely uncouple conformational coupling of CaV1.1 with RYR1 within the junction? Do molecular dynamic simulations of the VSDIII-LIV constructs reveal defects in normal vertical/lateral transitions of S4 VSD III? Could inclusion of LIV impair the junctional targeting or functional role of another essential protein of ECC (e.g. β 1a or Stac3). Do these constructs generate expected levels of charge movement (see comment 1 above)?
3. Given the objective of assessing the respective role of each VSD to CaV1.1 calcium channel and voltage sensor function, it is surprising that parallel studies in VSD IV were not completed. While the

authors have assessed the role CaV1.1 VSD IV in other studies, specific effects of neutralizing mutations to the innermost S4 VSD IV charged residues on the voltage dependence and kinetics of CaV1.1 calcium currents and calcium transients have not been reported. Inclusion of these experiments in VSD IV would complete the intended comprehensive analyses of Cav1.1.

4. It is not clear why the maximum magnitude of calcium release ($\Delta F/F_0$) for WT CaV1.1 shown in Figure 1 is $\sim 4\times$ smaller (0.5 vs 2.0) than that shown for this construct in all other figures.

5. The figures included in the manuscript are small and of relatively low resolution, making it difficult to make out individual labels. It would be helpful if the authors could provide larger figures at a much higher resolution.

6. Panel letters (A, B, and C) need to be added to Figure 6 to match the text and figure legend.

Reviewer #2 (Remarks to the Author):

The authors use a variety of techniques to gain insight on the CaV1.1 ion channels. The paper is very nice to read. As I am a theoretician, I will comment only on the MD simulations carried out here.

The authors should address the following issues:

1. The membrane voltage applied (from 583 to 600 mV) is much higher than the physiological ones. Thus, the finding cannot be related to physiological states of the channel. This should be stated in the text. In addition, the authors should comment of the eventual differences between the voltage applied experimentally in vitro and that imposed computationally.

2. The parameters used for the barostat and the thermostat should be added.

3. I appreciate the use of a sophisticated model for calcium as the multi-site model by Zhang et al., but little detail of what the calcium ions do in the simulation is given. In particular, do they permeate in the channel? Can one compare calcium currents with experimental data in vitro made in the same conditions?

POINT-TO-POINT RESPONSE TO REVIEWER COMMENTS

Reviewer #1 (Remarks to the Author):

Pelizzari et al describe experiments that evaluate the relative role of the four voltage sensor domains of Ca_v1.1 (VSD I, VSD II, VSD III, and VSD IV) on the calcium channel (pore opening, calcium current) and voltage sensor (triggering SR calcium release) functions of Ca_v1.1.

...

The manuscript could be improved by the authors addressing the following points.

1. Surprisingly, no charge movement data were included in this study. As the S4 (and S3-S4 linker) mutations used in this study are predicted to exhibit specific effects on a given VSD (e.g. Ca_v1.1e VSDIII-LIV mutant is assumed to specifically alter VSD III function), effects on charge movement magnitude and voltage dependence could be linked to the behavior of a specific VSD. It would be most informative to assess charge movements for non-conducting versions of VSD III mutants that result in >100 mV shift in the voltage dependence of calcium release (e.g. comparing Ca_v1.1nc_VSDIII-R4A with Ca_v1.1nc_VSDIII-LIV). Assuming the specificity assumption above applies, differences in the voltage-dependence of charge movement for these two constructs would reflect changes in the voltage-dependence of VSD III, which could then be correlated with the simultaneously recorded calcium transients. For example, how closely does the voltage dependence of Ca_v1.1 VSDIII charge movement reflect the voltage dependence (V_{0.5} and k) of calcium release.

Author's response: We fully agree with the reviewer's assessment that precise charge movement data could be "most informative" for a mechanistic explanation and interpretation of the current results. Therefore, we undertook great efforts to obtain charge movement recordings; however, unfortunately without success.

A first challenge arises from the fact that our mutations affect a single VSD, while the charge movement recordings are expected to arise from all four VSD. Thus, the recordings need to be sufficiently robust to observe fractional changes of gating currents. Secondly, the four VSDs of Ca_v1.1 are expected to move at different voltages and with different kinetics (Ref. #7). Thus, their individual contributions to the measured gating charges may vary. Finally, charge movement recordings in cultured myotubes are notoriously hampered by the expression of various intrinsic voltage-gated channels (e.g. T-type calcium channels, Na_v...), the currents of which interfere with gating charge recordings at negative voltages, and the gating currents of which might contribute to (pollute) the charge movement measured. To circumvent this latter problem, we also turned to HEK cells. Here the experiments and interpretation of data were complicated by the necessity to co-transfect at least two additional proteins (beta1a and STAC1) to achieve Ca_v1.1 membrane expression and by the lack of EC coupling to independently assess functional expression of the non-conducting channels. In addition, it is not clear whether the absence of RyR1 might alter the natural movements of Ca_v1.1 VSDs.

Consequently, after three months of trying, we have not succeeded to generate charge movement data at the quality necessary to address the questions raised by the reviewer. We are not giving up tackling this challenge for future studies. But at this point we cannot supply reliable charge movement data.

2. The authors do not discuss reasons for why the maximum magnitude of calcium release (ECC) is reduced by >90% for the two VSDIII-LIV constructs shown in Fig. 4O (CaV1.1e_VSDIII-LIV and CaV1.1nc_VSDIII-LIV)? This is one of the largest effects observed in this study, and surprisingly, is essentially ignored by the authors. While ICC results confirm normal junctional targeting of these constructs, could inclusion of LIV limit tetrad formation or otherwise largely uncouple conformational coupling of CaV1.1 with RYR1 within the junction? Do molecular dynamic simulations of the VSDIII-LIV constructs reveal defects in normal vertical/lateral transitions of S4 VSD III? Could inclusion of LIV impair the junctional targeting or functional role of another essential protein of ECC (e.g. β 1a or Stac3). Do these constructs generate expected levels of charge movement (see comment 1 above)?

Author's response: We thank the reviewer for noticing this important point. Our analysis of the cause of this reduced calcium release amplitude in the left-shifted chimeras (CaV1.1e_VSDIII-LIV and CaV1.1nc_VSDIII-LIV) quickly revealed that it was an artifact arising from our routinely used pre-pulse protocol to block potential interference of T-Type currents. In the strongly left-shifted chimeras, the pre-pulse to -30 mV activated SR calcium release. Consequently, at the start of the test pulse, the baseline was still elevated and DF/F of the calcium transients was curtailed. To prevent this artifact and to achieve comparability between the control and the left shifted chimeras, we repeated the full set of experiments shown in Fig. 4 I-P without the pre-pulse.

The central findings (i.e. the left-shifted voltage-dependence of EC coupling with CaV1.1e_VSDIII-LIV and CaV1.1nc_VSDIII-LIV while current properties of CaV1.1e_VSDIII-LIV were unaffected) have not changed at all. However, using the appropriately modified protocol the reduction of the EC coupling amplitude is much less and the mean is no longer significantly different from control (see new Fig 4 and corrected data in Table 1; revised text in lines 277-279).

3. Given the objective of assessing the respective role of each VSD to CaV1.1 calcium channel and voltage sensor function, it is surprising that parallel studies in VSD IV were not completed. While the authors have assessed the role CaV1.1 VSD IV in other studies, specific effects of neutralizing mutations to the innermost S4 VSD IV charged residues on the voltage dependence and kinetics of CaV1.1 calcium currents and calcium transients have not been reported. Inclusion of these experiments in VSD IV would complete the intended comprehensive analyses of Cav1.1.

Author's response: Initially we had not included this experiment because in earlier experiments we already demonstrated that the alternative splicing of exon 29 (insertion in the IVS3-S4 loop) right-shifted $V_{1/2}$ of current activation by 30 mV without affecting EC coupling (Ref. #16). However, we agree with the reviewer that inclusion of experiments neutralizing the innermost gating charges of VSD IV "would complete the intended comprehensive analysis of Ca_v1.1." Therefore, we generated the corresponding constructs in Ca_v1.1a and Ca_v1.1e and conducted these experiments.

The new data are now displayed in Fig. 2 I-P, Table 1, and Supplementary Fig. 1A; the additional results are described in lines 133-148 and the molecular biology in lines 474-480. Consistent with the overall results, the mutation altered the voltage-dependence of current gating in the adult Ca_v1.1a isoform, but had no effect on the voltage-dependence of EC coupling in either isoform. Moreover, this data set once more demonstrates that EC coupling

is also unaffected by the insertion of exon 29 in VSD IV, which right-shifted $V_{1/2}$ of current activation.

4. It is not clear why the maximum magnitude of calcium release ($\Delta F/F_0$) for WT CaV1.1 shown in Figure 1 is $\sim 4\times$ smaller (0.5 vs 2.0) than that shown for this construct in all other figures.

Author's response: The magnitude of the calcium transients can vary considerably depending on the quality of the myotubes, also resulting in long-term fluctuations of control values. Therefore, we always analyze/compare data obtained in matched experiments (same cell passage, same transfections). The particular data set of Fig. 2 A-H had been recorded quite some time before the rest of the experiments, at a period when cultures were less than optimal (during the pandemic period). While the calcium signals are rather small, this does in no way affect the validity of the result. Moreover, in the context of this article this experiment was intended as proof of principle confirming the earlier findings with R174W mutations (Ref. #20). Therefore, we did not redo these experiments even though the expression levels / amplitudes do not match those from later recordings.

5. The figures included in the manuscript are small and of relatively low resolution, making it difficult to make out individual labels. It would be helpful if the authors could provide larger figures at a much higher resolution.

Author's response: The figures in the combined PDF used in the review process were compressed to achieve a manageable file size. For final submission high-resolution figures will be uploaded which should result in the expected high quality.

6. Panel letters (A, B, and C) need to be added to Figure 6 to match the text and figure legend.

Author's response: Done as requested

Reviewer #2 (Remarks to the Author):

The authors use a variety of techniques to gain insight on the CaV1.1 ion channels. The paper is very nice to read. As I am a theoretician, I will comment only on the MD simulations carried out here.

The authors should address the following issues:

1. The membrane voltage applied (from 583 to 600 mV) is much higher than the physiological ones. Thus, the finding cannot be related to physiological states of the channel. This should be stated in the text. In addition, the authors should comment of the eventual differences between the voltage applied experimentally in vitro and that imposed computationally.

Author's response: Excessive potentials are necessary to achieve VSD state transitions within the $<5 \mu s$ simulation time. At physiological potentials the VSD state transitions are expected to occur at the physiological time scale of ms. However, ms-long MD simulations are not computationally feasible with contemporary computing power. Thus, the use of excessive

voltages is state-of-the-art in the field (<https://elifesciences.org/articles/53400#s4>). As requested by the reviewer, this fact is now stated in the text (lines 607-609).

Furthermore, within the voltage-sensing domain, the membrane electric field is concentrated at the narrow portion of the hydrophobic constriction site (Delemotte et al., PNAS, 2011). Therefore, the actual field strength in the hydrophobic constriction site might approach that used in the MD simulation.

In the case, the excessive electric potential distorted the protein structure, this becomes immediately apparent (e.g. collapse of helix). In the course of experimental optimization, such simulations are excluded and the parameters (i.e. electric field) adjusted to tolerated values.

2. The parameters used for the barostat and the thermostat should be added.

Author's response: As requested by the reviewer, we added the following paragraph in the Methods section (line 600-603): "The temperature was maintained at $T = 310$ K using the Nosé-Hoover thermostat with the inverse friction constant set to 1.0 ps^{54} , and the pressure was maintained semi-isotropically at 1 bar using the Parrinello-Rahman barostat with the period of pressure fluctuations at equilibrium set to 5.0 ps and the compressibility set to $4.5e-5 \text{ bar}^{-155}$."

3. I appreciate the use of a sophisticated model for calcium as the multi-site model by Zhang et al., but little detail of what the calcium ions do in the simulation is given. In particular, do they permeate in the channel? Can one compare calcium currents with experimental data in vitro made in the same conditions?

Author's response: In the current simulation $\text{Cav}1.1$ transitions from the inactivated state (VSDs up, pore closed) into the resting state (deactivating direction). Therefore, calcium permeation is not expected in any state and actually not observed. In an ongoing follow-up study, we utilize this MD model to simulate the state transitions in the activating direction. There, the question as to whether the pore opens and calcium permeates becomes highly relevant.

REVIEWERS' COMMENTS

Reviewer #1 (Remarks to the Author):

The revised manuscript is significantly improved.

Specifically, Pelizzari et al now include new experiments that evaluate the relative role of repeat IV voltage sensor domain of CaV1.1 (VSD IV) on the calcium channel (pore opening, calcium current) and voltage sensor (triggering SR calcium release) functions of CaV1.1. The results of these studies confirm the authors overall conclusion that while all 4 VSDs differentially contribute to CaV1.1 calcium channel function, only VSD III controls CaV1.1 voltage sensor function (i.e. RyR1 calcium release during ECC). As a result, the findings reported in the revised manuscript now provide a more comprehensive assessment of the impact of all four repeats on CaV1.1 channel and voltage sensor function.

The authors attempted to complete the requested charge movement measurements for a non-conducting version of a VSD III mutant that result in >100 mV shift in the voltage dependence of calcium release (e.g. comparing CaV1.1nc_VSDIII-R4A with CaV1.1nc_VSDIII-LIV). Unfortunately, for technical and resolution issues, sufficiently reliable data for these studies could not be generated. The authors are encouraged to consider the use of voltage clamp fluorometry to resolve depolarization-induced structural changes of individual CaV1.1 VSDs, which was successfully implemented for CaV1.1 expressed in xenopus oocytes (PMID 34546289), though admittedly, translation of this approach to myotubes could be challenging. However, even expression studies in Xenopus oocytes could be used to specifically compare the voltage-dependence of VSD III charge movement for the two above constructs with very different voltage-dependences for Ca release already documented in myotubes.

In their response letter, the authors provide reasonable explanations for all other comments and concerns raised in the prior review including reasons for the maximum magnitude of calcium release being reduced by >90% for the two VSDIII-LIV constructs (pre-pulse effect corrected by conducting new experiments without a no pre-pulse) and why the maximum magnitude of calcium release for WT CaV1.1 in Figure 2 is ~4x smaller than that shown in other figures (early experimental results differing quantitatively from later experiments with the same construct conducted with healthier myotubes).

I could not find a description of the pre-pulse protocol used in the Methods section of the manuscript, This was likely an oversight. Thus, the Methods section of the manuscript should be revised to include a description of the pre-pulse protocol used to inactivate T-type calcium channels (1-sec to -30 mV) and the constructs for which this pre-pulse protocol was not used (and why).

Reviewer #2 (Remarks to the Author):

I appreciate that the authors now state that they use unphysiologically high potentials in the text. However, the authors should add what this implies when comparing their results with those obtained by experiment, which might have been carried out instead in physiological conditions.

POINT-TO-POINT RESPONSE TO REVIEWER COMMENTS

Reviewer #1 (Remarks to the Author):

The revised manuscript is significantly improved.

Specifically, Pelizzari et al now include new experiments ... As a result, the findings reported in the revised manuscript now provide a more comprehensive assessment of the impact of all four repeats on CaV1.1 channel and voltage sensor function.

Author's response: Thanks for noting!

The authors attempted to complete the requested charge movement measurements ... The authors are encouraged to consider the use of voltage clamp fluorometry ... though admittedly, translation of this approach to myotubes could be challenging. ...

Author's response: We fully agree with the reviewer's assessment that, although challenging, voltage clamp fluorometry (if not in muscle than in oocytes) is the way to go. We are "encouraged" to pursue this avenue in our continuing efforts to better understand the voltage sensing of EC coupling.

In their response letter, the authors provide reasonable explanations for all other comments and concerns raised in the prior review ...

Author's response: Thanks for noting!

I could not find a description of the pre-pulse protocol used in the Methods section of the manuscript, This was likely an oversight. Thus, the Methods section of the manuscript should be revised to include a description of the pre-pulse protocol used to inactivate T-type calcium channels (1-sec to -30 mV) and the constructs for which this pre-pulse protocol was not used (and why).

Author's response: As requested by the reviewer, we included the pre-pulse protocol in the Materials and Methods of the final version of the manuscript.

Reviewer #2 (Remarks to the Author):

I appreciate that the authors now state that they use unphysiologically high potentials in the text. However, the authors should add what this implies when comparing their results with those obtained by experiment, which might have been carried out instead in physiological conditions.

Author's response: We already had stated in the revised manuscript that the use of high voltage was necessary to overcome the time scale difference between physiological channel activation and observing the state transitions in MD simulations. As requested by the reviewer, now we further added a statement concerning the validity of the approach and its consequences on the resulting resting state structures in comparison to experimental structures of related channels.